# Diet and Physical Activity in Adult Dominant Polycystic Kidney Disease: A Review of the Literature

**DOI:** 10.3390/nu15112621

**Published:** 2023-06-03

**Authors:** Irene Capelli, Sarah Lerario, Valeria Aiello, Michele Provenzano, Roberta Di Costanzo, Andrea Squadrani, Anna Vella, Valentina Vicennati, Carolina Poli, Gaetano La Manna, Olga Baraldi

**Affiliations:** 1Nephrology, Dialysis and Renal Transplant Unit, IRCCS Azienda Ospedaliero-Universitaria di Bologna, 40138 Bologna, Italy; irene.capelli4@unibo.it (I.C.); sarah.lerario2@unibo.it (S.L.); valeria.aiello6@unibo.it (V.A.); michele.provenzano2@unibo.it (M.P.); roberta.dicostanzo@studio.unibo.it (R.D.C.); anna.vella4@studio.unibo.it (A.V.); olga.baraldi@aosp.bo.it (O.B.); 2Department of Medical and Surgical Sciences (DIMEC), Alma Mater Studiorum University of Bologna, 40138 Bologna, Italy; andrea.squadrani4@studio.unibo.it (A.S.); valentina.vicennati2@unibo.it (V.V.); carolina.poli4@unibo.it (C.P.); 3Division of Endocrinology and Diabetes Prevention and Care, IRCCS Azienda Ospedaliero-Universitaria di Bologna, 40138 Bologna, Italy; 4Sviluppo Professionale e Implementazione della Ricerca nelle Professioni Sanitarie, IRCCS Azienda Ospedaliero-Universitaria di Bologna, 40138 Bologna, Italy

**Keywords:** polycystic kidney disease, ADPKD, nutrition, ketogenic diet, physical activity

## Abstract

Autosomal polycystic kidney disease is the most common inherited kidney disease determining 5% of all end-stage kidney disease. The only therapy approved for this condition is Tolvaptan, which, with its aquaretic effect, has a strong effect on patients’ daily life. Recently, the literature has been enriched with new works that analyze possible non-pharmacological therapeutic strategies to slow cysts’ enlargement and chronic kidney disease progression. Among them, dietary schemes reducing carbohydrate intake and inducing ketoses have been demonstrated to have efficacy in several pre-clinical and clinical studies. A ketogenic diet, calorie restriction, intermittent fasting, and time-restricted feeding can reduce aerobic glycolysis and inhibit the mTOR pathway, producing a reduction in cyst cell proliferation, a reduction in kidney volume, and helping to preserve kidney function. ADPKD’s burden of disease has an impact on patients’ quality of life, and the possibility to play sports or carry out physical exercise can help people in everyday life. The multisystemic character of the disease, especially cardiovascular involvement, needs to be carefully evaluated to establish the quality and quantity of physical activity that patients can safely carry out.

## 1. Introduction

Autosomal polycystic kidney disease (ADPKD) is the most common inherited kidney disease and is characterized by a gradual and slow formation and growth of kidney cysts leading to end-stage kidney disease. In recent years, the lifespan of polycystic patients has increased thanks to several strategies that have improved the symptoms and complications of the disease. Tolvaptan is the only drug approved for ADPKD treatment and it has been demonstrated to be effective in the reduction of cyst growth, but its aquaretic effect may change patients’ daily living habits.

An Increasing number of studies report important association between diet, physical activity (PA), and health-related quality of life (QoL) both in the general population and in several pathological contexts, such as in oncological patients [1,2,3]. At the same time, in the literature, there has been growing interest in patients’ points of view on pathologies, treatment, therapeutic programs, and their impact on everyday life. In this regard, PROMs (patient-reported outcome measures) have been developed and are increasingly used to obtain feedback from the patient, aiming to improve the quality of care.

Some studies have recently evaluated the impact of diet on the disease and have also hypothesized the use of dietary strategies for the treatment of the disease. The purpose of this review is to describe the specific domains of diet and physical activity for polycystic disease, laying the basis for a complete overview of these areas which widely influence the life of polycystic patients, and which must be taken into strong consideration during their multispecialty management.

## 2. Epidemiology and Pathogenesis of ADPKD

The incidence of ADPKD is about 1/400–1/1000 born alive [4], with an estimated prevalence in Europe of 329 subjects in a million. These numbers make it the most common hereditary kidney genetic disorder, associated with almost 5% of all terminal chronic kidney insufficiency (ESKD) [5].

Mutations in two genes, PKD1 (chromosome 16p13.3) and PKD2 (chromosome 4q21), are responsible for the disease, in particular, changes in PKD1 are attributable to 85% of cases while most of the remaining cases (10%) come from alterations in PKD2 [6]. ADPKD is characterized by high phenotypic variability, in particular, mutations in PKD2 lead to less aggressive disease, with end-stage kidney disease (ESKD) which occurs at the mean age of 79.7 years versus 58.1 in PKD1 mutation. In addition, the mutation of PDK1 itself shows a variety of pathological findings that are more severe with truncating mutations [7]. Nevertheless, extreme variability in presentation is well described also between subjects of the same family, suggesting that several factors may contribute to determining the course and the severity of the disease. Among these factors are additional mutations in modifier genes involved in the signal translation pathway (COL4A1 and HNFB1B); kidney damage from ischemia–reperfusion and tubular obstruction from crystal deposition; concomitant clinical conditions, including obesity, diabetes, vascular disease, and acute kidney damage; and lifestyle factors such as diet, water intake, and smoking habits. All these aspects may contribute to the cystogenesis and progression of ADPKD in different ways among patients [8,9].

PKD1 and PKD2 encode two integral membrane proteins associated with the primary cilia called polycystin 1 (PC1) and polycystin 2 (PC2), respectively. PC1 has a role in cell–cell interaction and matrix–cell interaction, it has a receptor function, modulating the opening of voltage-dependent calcium channels and potassium channels. PC2 is instead a voltage-dependent calcium channel [7]. The interaction of PC1 and PC2 forms a complex on the primary cilia, responsible for intracellular regulation of calcium [10]. Several hypotheses have been made about cystogenesis in ADPKD, but the mechanism according to which the reduction of signaling of polycystin determines polycystic disease remains largely unknown. According to the second-hit model, cystogenesis begins when mutations occur in the normal allele, increasing the loss of functionality of polycystins from 50% to 100%. However, recent evidence suggests that it is not necessary to have a complete loss of functionality of PC1 and PC2, for this to occur, whereas it is sufficient to have a reduction below a threshold beyond which the dosage of functioning polycystin correlates inversely with the severity of the disease [7].

Studies have shown that the flexion of the primary cilia produced by kidney flow is transduced through changes in intracellular calcium concentration [11]. According to these studies, the nephron’s primary would act as a sensor of flow reduction. When this condition occurs, the transduction of the primary cilia signal leads to the activation of PC1, which triggers a chemical response by activating PC2. The latter would mediate the entry of extracellular calcium into the cytosol which also triggers the release of calcium from intra-cellular deposits by activating the mTOR (mammalian target of rapamycin) signaling pathway. PC1 interacts with tuberin and mTOR, forming a complex with negative regulatory activity on mTOR itself. This and other signals trigger a response in tubular epithelial cells which involves de-differentiation and cell proliferation. In ADPKD, this program is continuously activated due to ciliary dysfunction, thus leading to interstitial fibrosis and cyst growth. Cells with mutated PKD1 or PKD2 genes appear normal but are insensitive to the flow into the nephron; a lack of functioning polycystin 1 with calcium intake leads to a cellular response that simulates a permanently activated primary cilia signal leading to tissue remodeling and a compensatory increase in tubules’ diameters, as well as other ciliary ducts such as those of the bile and pancreas [10].

The role of cAMP (cyclic adenosine monophosphate) is crucial; mutations in PKD1 and PKD2 lead to a reduction in intracellular calcium and an increase in cAMP. This activates protein kinase A (PKA) and increases the sensitivity of the principal cells of the collector duct to the constant stimulus of vasopressin (AVP); it promotes altered tubulogenesis, cell proliferation, increased secretion of fluids, and interstitial inflammation [7]. AVP also stimulates a further increase in cAMP levels by binding to V2 receptors.

Currently, Tolvaptan, a highly selective antagonist at the V2 vasopressin receptor (AVP), is a specific therapy approved for the treatment of ADPKD; it has been demonstrated that Tolvaptan reduces the growth rate of cysts and slows the increase in TKV, slowing ESKD. As described earlier, AVP binding with its V2 receptors stimulates cAMP production in the distal nephron and collector duct inducing cyst growth. The effects achieved by blocking AVP action through an antagonist are aquaresis, reduction in urinary osmolarity, and reduction in cyst proliferation mediated by the suppression of cAMP production [6,12].

## 3. Diet in ADPKD

Dietary interventions are a fundamental part of chronic kidney disease (CKD) treatment, demonstrated by their impact on slowing the progression of CKD and reducing the accumulation of metabolic products, helping in lowering symptoms of uremia and metabolic acidosis and lowering phosphate levels.

Among the dietary regimens, plant-based diets and dietary approaches to stop hypertension (DASH) may be beneficial in slowing CKD progression with their low sodium, saturated fat, phosphate apport, and high fiber intake. In addition, plant-based regimens are alkaline-forming, helping in the reduction of acidosis condition in advanced chronic kidney disease [13,14,15,16].

Similarly, the Mediterranean diet is advantageous in slowing kidney damage progression and cardiovascular disease thanks to the reduction in oxidative stress [17].

However, to the best of our knowledge, there are no studies published in the literature that analyze the efficacy of these regimens in reducing cyst formation and growth in ADPKD patients, even if it could be an interesting field to examine because of the possible high adherence to this diet.

In ADPKD patients, nutrition must be appropriate to the management of CKD, aiming to limit its complications and the progression toward end-stage kidney disease, while maintaining a good nutritional status [18]. Moreover, several studies have recently evaluated the impact of different nutritional strategies on cystogenesis and therefore on the progression of the specific damage of the disease (Figure 1). We have therefore dealt with these two different aspects related to diet in polycystic disease below.

### 3.1. Nutritional Aspects for the Management of CKD in ADPKD Patients

Most of the nutritional suggestions for the management of CKD in ADPKD derive from the general recommendations for CKD due to different etiology in particular.

#### 3.1.1. Proteins

Proper protein intake can help prevent or improve part of the metabolic imbalances due to chronic kidney disease. Prescription of a low-protein dietary regime remains one of the most widely used strategies for the management of symptoms related to uremic toxicity in CKD patients. The amount of protein is defined by the CKD stage: (Table 1) stages 1, 2, and 3a 0.8 g/kg/day; at stage 3b, protein restriction is 0.6 g/kg/day; at stages 4 and 5, this is 0.6 g/kg/day or 0.4–0.3 g/kg/day with amino acids or ketoanalogues supplementation [19].

Protein restriction has been proven to delay evolution to ESKD in non-ADPKD CKD patients, as shown in the Modification of Diet in Renal Disease (MDRD) study [20] where it was demonstrated that a low protein diet results in a slower decline in glomerular filtration rate (GFR). Studies in ADPKD patients have failed to demonstrate the slowing of CKD progression, as shown in the analysis of data from the ADPKD cohort of the MDRD study, Kramers et al.’s study, and the CRISP study [20,21,22].

The CRISP study initially showed an association between total kidney function increase and GFR decline and protein intake; however this association was not confirmed after adjustment for important confounders [22].

An elegant work conducted by Heida et al. showed a strong association between a high urine-to-plasma urea ratio and GFR decline in polycystic adult patients. Considering that the urine-to-plasma urea ratio positively correlates with protein intake, such a result suggests how low protein consumption could slow the worsening of kidney function even in ADPKD patients [23].

However, the majority of experts suggest the avoidance of very low protein intake (<0.6 g/kg/die) for malnutrition risk and suggest that if a similar regimen is adopted there be strict and careful monitoring of patients with trimestral or at least semestral follow-up [24].

Due to the lack of strong evidence, there is not a widely accepted guideline on protein intake in ADPKD patients and neither does the KDIGO Controversies Conference on ADPKD specify a protein regime [4]. The KDOQI guidelines for nutrition in CKD recommend protein restriction but without a specific evaluation of polycystic patients [4].

#### 3.1.2. Sodium

The importance of lowering salt intake is a priority in CKD patients. The aim is to keep sodium intake lower, to 90 mmol/day, corresponding to the consumption of less than 2000 mg/day of sodium (Na) and <5 g/day of sodium chloride (NaCl). This leads to better management of blood pressure, limitation of kidney damage, and a better cardiovascular outcome [25].

#### 3.1.3. Phosphorus and Potassium

Phosphate homeostasis abnormalities are frequently observed in patients with reduced kidney function and are often predictive of negative clinical outcomes. Hyperphosphatemia contributes to the burden of cardiovascular disease leading to cardiovascular and soft-tissue calcifications, cardiac disease, renal osteodystrophy, and secondary hyperparathyroidism. Phosphorus requirements depend on the stage of CKD and should consider the avoidance of malnutrition, not unusual in ESKD and dialysis patients [26].

Kidney’s potassium excretion is reduced with the lowering of the glomerular filtration rate, and hyperkalemia is a severe condition commonly experienced by CKD patients. Monitoring of potassium serum levels is fundamental, and the target values are 3.5–5.0 mmol/L to be reached with lower potassium diets and, if necessary, with hypokalemic drugs [26].

### 3.2. Nutritional Aspects with Possible Impact on the Pathogenesis of ADPKD

#### 3.2.1. Proteins

High protein intake is associated with hyperfiltration secondary to the secretion of glucagon, strictly associated with gluconeogenesis in the liver and with urea excretion by the kidneys, together with cAMP released from the liver. A protein-rich meal induces simultaneous vasopressin release improving urine concentrating ability, as shown in some studies on healthy humans in which GFR after high protein intake significantly correlates with increasing urine osmolality. The two hormonal pathways lead to increased urea excretion and water reabsorption, reducing the tubuloglomerular feedback (TGF) signal at macula densa and then increasing GFR, inducing hyperfiltration [27]. This mechanism, linked to vasopressin release, can at least partially explain the reduction in cystogenesis observed by reducing protein intake.

Indeed, in a study on DBA/2FG-pcy (pcy) mice, Aukema et al. showed how a low protein regimen has a positive impact in reducing cyst growth and kidney volume in comparison with a normal protein diet [28]. The same authors also demonstrated that dietary protein source and sex also affect size and fluid content in polycystic kidneys. In this study, the pcy mice were divided into two groups, a low protein regimen and a normal protein regimen; both groups were divided into two subgroups. In one, the animals were fed with soy protein isolate and in the other, the animals were fed with casein. The results show that the kidney weights were 28% lower in the animals fed with the low-protein regime based on soy compared to the mice consuming the casein-based diet, as the area of the parenchyma occupied by cysts was also 19% lower in the same group [29].

#### 3.2.2. Water Intake

Vasopressin (AVP) stimulates cyst growth in ADPKD. Increased water intake is recommended for AVP secretion suppression, which reduces AMPc synthesis and consequently cyst growth [7] and improves kidney function, as shown in experiments on ADPKD mice [30,31] 

The DRINK study explored a high-water intake regimen in humans as an alternative to pharmacological therapy in blocking vasopressin. The patients were divided into two groups: high water intake (HW) and ad libitum intake (AW); the 8-week experiment showed that the osmolarity target (<270 mOsm/kg) could be reached by the majority of the patients in the HW intake group without difficulty, demonstrating the feasibility, safety, and adherence to a possible water therapy [32].

Another study published by Rangan et al. in 2022 studied two groups of ADPKD patients, one with high water intake versus one with ad libitum intake, and even though a reduction in urinary osmolarity was observed in the high-intake group, no differences in kidney volume emerged [33].

What is important to underline is the necessity of further trials to verify the effects of water intake on the progression of ADPKD.

Otherwise, increased water intake has also been shown to counteract nephrolithiasis, reducing the risk of stone formation, one of the main comorbidities of ADPKD [34].

#### 3.2.3. Sodium

Sodium restriction in ADPKD may be favorable because of the reduction in renin angiotensin aldosterone system activity. Aldosterone induces the transcription of the alpha subunit of Na/K ATPase which may promote, in ADPKD patients, fluid secretion and cyst enlargement. In addition, through the epidermal growth factor receptor, it stimulates kidney fibroblast proliferation and the proliferation of dedifferentiated kidney tubule cells [35].

Additionally, blood pressure control has been demonstrated to be fundamental in disease progression. As with the other causes of CKD, even in ADPKD patients, strict management leads to a slower reduction in GFR and a lower annual increase in kidney volume [36].

T restriction also reduces vasopressin levels. Amro et al. [37] performed a two-week randomized trial to determine whether a low-sodium (1500 mg/day) and normoproteic (0.8 g/kg) diet, with adjusted water intake, would suppress AVP secretion in 34 ADPKD patients. The patients were randomly divided into intervention and control groups, and plasma copeptin and urinary osmolarity levels were used as indicators of AVP suppression. At the end of the observation period, there was a significant decrease in the intervention group for both values: copeptin decreased from 6.2 ± 3.05 to 5.3 ± 2.5 pmol/L (*p* = 0.02), and urinary osmolarity decreased from 426 ± 193 to 258 ± 117 mosm/kg (*p* = 0.01), while they remained unchanged in the control group. Thus, suppression of AVP is also possible by managing sodium intake and without large increases in water intake, which may be more difficult to maintain in the long term.

Furthermore, in Hane:SPRD mice (a model for polycystic kidney disease), an increase in sodium intake is accompanied by a higher kidney weight and cyst dimension [38]. 

However, the literature on randomized trials investigating the effect of a reduction in salt intake in the progression of ADPKD disease is scarce. The best evidence available is from a post hoc analysis of the HALT A and B trials where the effect of blood pressure control and RAAS blockade was investigated in ADPKD. In particular, the post hoc analysis showed that a decrease of 1 g in salt intake is associated with a reduction in kidney growth (0.43%/year) and GFR decline (−0.09 mL/min/1.73 m^2^/year for each gram of salt reduction) and also with a lower incidence of composite renal endpoint (reduction of eGFR of 50%, ESKD or death) [39]. In conclusion, the data suggest a beneficial effect of sodium restriction in the management of ADPKD.

#### 3.2.4. Phosphorus

High phosphate levels in ADPKD could induce tubular injury, promoting cyst growth and disease progression.

Omede et al. suggested that restriction in the dietary intake of phosphates may be able to slow cystic development and fibrosis in PKD. To verify this, the authors divided the PKD mice into two identical groups and fed them with two regimes differing only in phosphate intake (normo-phosphoric and hypo-phosphoric). Dietary phosphate restriction resulted in a 25% lower kidney weight/body weight ratio, a reduction in the number of cysts, and expression of tubular injury markers (neutrophil gelatinase-associated lipocalin—NGAL) [40].

#### 3.2.5. Caffeine

Caffeine is a methylxanthine that increases the levels of intracellular cAMP in cultured renal ADPKD epithelial cells. A retrospective analysis of the CRISP study cohort found no association between caffeine consumption and a higher increase in hTKV (height-adjusted total kidney volume), progression to ESKD, or death in 239 patients with an average follow-up of 12.5 years [41]. Even data from a prospective longitudinal study that analyzed caffeine consumption and kidney volume and function did not find an association [42]. Secondary to the absence of strong evidence, the limitation in caffeine consumption in ADPKD patients is limited to general suggestions [4].

#### 3.2.6. Calories and Body Weight

Appropriate nutrients and calorie intake must always be reached in order to avoid malnutrition and related complications [18]. Caloric restriction under the supervision of doctors and dietitians is desirable if weight loss is necessary for either the general population or in CKD patients.

There are currently few studies concerning the association between overweight/obesity and the decline of kidney function in ADPKD. Nowak et al. investigated this correlation by analyzing data from 441 non-diabetic participants in the HALT PKD study A with early ADPKD [43]. The subjects were divided into normal weight, overweight, and obese based on BMI (body mass index) calculated excluding the weight of the kidneys and liver. A higher BMI was associated with a greater increase in TKV over time (normal weight: 6.1% ± 4.7%, overweight: 7.9% ± 4.8%, and obese: 9.4% ± 6.2%; *p* < 0.001) after a follow-up of five years. This could be due to continuous caloric excess, leading to strong activation of the mTOR pathway with relative suppression of AMPK (adenosine monophosphate-activated protein kinase), leading to the proliferation of cysts. Following this hypothesis, it has been thought that slight/moderate calorie restriction, activating AMPK and suppressing mTOR, could slow the progression of the disease [44]. This has been verified in animal model experiments. For example, Kipp et al. [45] showed how, in an orthologous mouse model of ADPKD, a reduction in food intake by 23% slows disease progression without affecting body weight or causing malnutrition or any side effects. An interesting and elegant experiment conducted by Hopp and colleagues [46] investigated the feasibility of daily caloric restriction (DCR) and intermittent fasting (IMF) in overweight ADPKD patients, showing significant weight loss and a reduction in cyst growth even with DCR and IMF, although DCR was more tolerated. In parallel, they compared the efficacy of DCR, IMF, and time-restricted feeding (TRF) in the ADPKD mouse model and they found that only mice on DCR had significant weight loss and slowing down of cyst growth. Even this work confirms the benefit of weight loss in ADPKD progression on human and on mice and emphasizes the feasibility of a daily caloric restriction regimen.

The beneficial effects of food restriction in several aspects of health appears to be related to the insulin growth factor-1 (IGF-1) pathway, which is also dysregulated in cancer and aging. In ADPKD, IGF-1 could be involved in cyst proliferation; in fact, some studies have found increased expression of IGF-1 in polycystic patients and animal models [47]. The work of S. Kashyap et al. [48] identified PAPP-A, a metalloprotease that cleaves IGF-1 binding protein and frees IFG-1, as a crucial factor in increasing IGF-1 bioavailability and which opens new future research for understanding ADPKD pathological mechanisms and for discovering new therapeutic targets.

#### 3.2.7. Ketogenic Diet

In addition to the classic diet approach, interesting new strategies are being increasingly studied, such as calorie restriction or diets inducing ketosis states.

A KD is carried out with careful food planning, with macronutrients in predetermined proportions. The classic approach is based on the lipid ratio which can vary from a four parts fat (long-chain triglycerides) to a one-part combined protein and carbohydrate ratio (from 4:1 to 2:1), supplemented with minerals and vitamins [49].

Nowadays, in addition to the classic KD mentioned above, there are new approaches able to induce a ketogenic status, more tolerated in the long-term, such as the modified Atkins diet, the MCT diet, or versions of intermittent fasting (for example time-restricted feeding), which are commonly used in daily practice.

Recent studies [45,50,51,52] have shown that a diet able to induce moderate calorie restriction, such as time-restricted feeding (TRF) or classical intermittent fasting, may be effective in counteracting cyst growth. The first experiment was carried out on animal models of PKD; the results showed that a moderate calorie restriction regime led to a reduction in the size and growth rate of cysts and inflammation levels. The mechanism responsible for the positive effect of the restriction is thought to be the inhibition of the mTOR transduction pathway, which in ADPKD is pathologically hyperactive, leading to cyst growth and forced inhibition of AMPK. Under calorie restriction conditions, the ATP/AMP ratio decreases, leading to activation of AMPK, inhibition of mTOR, and then to a correct balance between AMPK and mTOR. 

The rationale behind using a KD approach in ADPKD patients has been explored by several works [45,53,54,55]. Cyst cells’ metabolism is similar to cancer cells’ one in terms of energy production, and with a mechanism comparable to the Warburg effect, they rely almost exclusively on glycolysis to produce energy. For this reason, it has been hypothesized that a ketosis state with a dramatic reduction in glucose availability, an increase in fatty acid (FFA) degradation, and the utilization of ketone bodies to produce energy can deprive cells of their main source of nutrition. Therefore, forcing tissue to use acetone, acetoacetic acid, and beta-hydroxybutyrate (BHB) induces a state of nutrient deficiency leading to mTOR inhibition and activation of AMPK and slowing cyst growth. Moreover, the inability of these cells to properly metabolize FFA could lead to excessive accumulation of lipid droplets at the cytoplasmic level, resulting in cell death due to lipotoxicity.

The positive effects of ketosis appear to be, at least in part, due to the increase in BHB, since this would be able to independently modulate the pathways of mTOR and AMPK. In fact, it has been demonstrated in animal models that using oral supplements containing BHB simulates the effects of ketosis induced by diet [56]. Torres et al. also state that the benefits induced by a regimen of TRF or classical intermittent fasting are secondary to ketosis since transient ketosis with a relative increase in BHB occurs in fasting [53,54]. In general, animal model studies suggest that a KD regimen with or without BHB supplementation may be an interesting option in the future of ADPKD therapies.

Moving from animal models to human evidence, Strubl et al. published a retrospective study in which they collected through questionnaires the experience of ADPKD patients who had already carried out ketogenic diets and TRDs (restricted-time diets) in the past in order to obtain information on the safety, feasibility and possible benefits of this regimen on the PKD patient [57]. From the initial candidates, once excluding all those who were on dialysis, those who had undergone kidney transplantation, or those with non-KD friendly diets, 131 participants remained, of whom 74 adopted a KD regime and 52 adopted a TRD for at least six months. The results show how most of the participants report an overall improvement in their quality of life and well-being, with a decrease in disease-related pain (hip and/or back pain, abdominal swelling, and early filling), a mean weight loss of 9.1 kg (greater in those who have adopted regimes of KD vs. TRD), and better pressure control. In addition, 45 participants also experienced a slight improvement in their glomerular filtration rate, with a mean increase of 3.6 mL/min/1.73 m^2^. The participants with documented ketosis showed the highest augmentation of eGFR, with an increase of 7.3 mL/min/1.73 m^2^. However, it is important to point out that the improvement in kidney function should be analyzed deeper with other techniques that are more accurate, such as creatinine clearance or kidney scintigraphy to define if it is real or is due to hyperfiltration or malnourishment which are not a desirable effect of the ketogenic regime. The principal side effects of the diet were fatigue, reported by 66% of participants, hunger, and the typical symptoms of keto-flu (headache, nausea, blurred mind, and weakness), symptoms that in almost all patients disappeared after a short time after adaptation to ketosis. Ultimately, most of the participants considered KD and TRF feasible; however, they reported a higher difficulty in long-term maintenance especially in the DK regimen. These results are promising, nevertheless, all data are reported by the patients themselves, obtained in an uncontrolled environment and without balance between men and women; thus, the reliability is very limited, with a strong bias, but this experience may be useful for future clinical trials.

The potential benefits of ketosis in ADPKD have recently been discovered and for this reason, there are very few clinical trials published or in progress concerning the ketogenic regimen in humans [58,59,60] and they are all preliminary studies with an extremely small cohort of patients. One of these is the Ren.nu program published by Bruen et al. It is a 16-week duration protocol with the aim of educating patients, through interviews with dietitians, to adopt and maintain a modified and plant-focused ketogenic diet [59]. The energy needs were calculated patient-based, with the macronutrients divided into 10–15% net CHO (total carbohydrates minus fibers), 10–15% PRO (protein), and 70–75% FAT (lipid), which avoids the excess of oxalates, phosphates, and uric acid and includes the use of a special supplement containing mainly BHB and citrates [59]. In addition to nutrition management, the patients were also trained to control the status of ketosis through monitoring their serum ketone levels. A first beta test phase with 24 selected participants ended in 2021; 20 participants completed the test and submitted questionnaires verifying the feasibility, tolerance, and adherence to the program. The results show improvements in terms of pain and general fatigue related to the disease, with good satisfaction regarding nutrition. The main issues were the side effects of ketosis and difficulty with eating in restaurants. Finally, the anthropometric measures, blood tests, and kidney function values were analyzed at the end of the experiment and compared with the basal values. The results showed an average weight drop of 5.6% (−4 kg), an average decrease in fasting blood sugar of 16.5% (−19 mg/dL), and a mean creatinine drop of 5.8% (−0.1 mg/dL) with an eGFR increase of 8.6% (+4.4 mL/min/1.73 m^2^). As mentioned before, even this work has the limitation of evaluating kidney function with the measure of creatinine and estimation of the glomerular filtration rate.

A study by Testa et al. tested the feasibility and potential benefits of a modified Atkins diet in ADPKD patients [60]. The choice of this scheme is due to its greater flexibility, which is why it is thought to be more applicable in the long term. The study lasted three months, three patients were enrolled, and macronutrients were calculated and divided based on the energy needs of patients with 5% CHO, 30% PRO, and 65% FAT. At the end of the experiment, the subjects completed questionnaires that revealed high satisfaction with the regimen and strong compliance, with only a few problems, concerning mainly the difficulties at restaurants and side effects due to the induction of the ketosis state. They observed an average increase in total cholesterol of 34 ± 13.1 mg/dL but it is not specified if the regimen prescribed was with specific ketogenic bar products or with fresh food, which could explain the increase in the cholesterol values. On the other hand, they found a decrease in blood sugar from 105.8 ± 8.5 mg/dL to 92 ± 8.8 mg/dL, and overweight patients (two of the three involved in the study), who were given a slightly low-calorie protocol, obtained a weight loss of 4.2 kg for one patient and 1 kg for the other. There was no change in kidney function.

Another interesting study, the first prospective interventional trial, RESET-PKD [61], confirms that KD induces ketogenesis which leads to a rapid change in total liver volume without modification to TKV. The authors suggest that the lack of response in kidney volume was secondary to the short duration of the intervention.

New randomized controlled clinical trials are currently ongoing and are evaluating time-restricted feeding [62], calorie restriction [63], and ketogenic dietary intervention [64] and their effect on slowing ADPKD progression, hoping for results that will allow for an increase in the number of therapeutic options for polycystic disease.

## 4. Physical Activity

Physical exercise in ADPKD patients has limited published data. In general, it is known how fundamental physical activity (PA) is in maintaining and improving cardiovascular and metabolic fitness, and the importance of controlling these factors in CKD is also known. It is reasonable to think that such benefits could also apply to ADPKD patients, where the burden of cardiovascular disease is often particularly significant.

The KDIGO 2021 guidelines recommend that all patients with kidney disease must be involved in at least 150 min of PA per week. The form and intensity of exercise should be constructed on an individual basis, as there may be health benefits even when levels of exercise are lower than recommended [65]. Physical exercise and sports contribute to lowering blood pressure and cholesterol, help to prevent becoming overweight and obese, fight diabetes and osteoporosis, and also positively affect psychological well-being, reducing anxiety, depression, and a sense of loneliness [66,67].

The condition of ADPKD requires some specific precautions in prescribing physical activity. In a healthy subject, physical exercise determines an increase in heart rate and output with simultaneous dilation of the peripheral vascular bed and a subsequent reduction in resistance; the increase in output raises systolic pressure, while diastolic pressure remains unchanged or decreases through peripheral vasodilation [67]. Several studies [68,69] have documented how subjects with ADPKD show an altered cardiovascular physiopathology both at the basal level and during exercise, although still without hypertension and with normal levels of GFR.

Martinez-Vea et al. studied young ADPKD patients after physical exercise to assess their cardiovascular response and they found echocardiographic signs of early diastolic dysfunction with an exalted response of systolic pressure and little decrease in diastolic pressure secondary to reduced vasodilating capacity [69]. Another characteristic of these patients seems to be an exalted response of the sympathetic autonomous system; the increase in heart rate normally stimulated by PA does not rapidly decrease to previous levels but persists longer. The heart rate recovery (HRR) index is a strong predictor of cardiovascular events when present with abnormal autonomic control of the heart and vasculature [70]. Finally, ADPKD patients present early arterial stiffness with the resulting increase in pulse wave velocity, which appears to be related to a latent systemic inflammatory condition [67].

Reinecke et al. evaluated the response to exercise in polycystic adults with no evidence of ventricular hypertrophy and with normal kidney function. The test consisted of 20 min of use of an exercise bike and bio-humoral measurement; the results were a reduced anaerobic threshold and reduced maximum oxygen consumption, with a low response of NO (nitric oxide) and ADMA (asymmetric dimethylarginine) during acute exercise. These are all indicators of the possible presence of early endothelial dysfunction in this pathology while there was no correlation between this type of cardiovascular abnormalities and TKV [68].

In conclusion, there is no doubt that patients affected by ADPKD, like other nephropathic patients, must perform regular exercise according to indications of KDIGO 2021 but with special precautions. For example, contact sports such as rugby, boxing, and football should be avoided in order to prevent abdominal trauma which could cause cysts to rupture and subsequent bleeding. Considering the cardiovascular profile previously described, it is recommended to carry out a complete cardiological screening before starting sports and exercises.

## 5. Conclusions

From the literature, it has emerged that the current nutritional approach towards patients with ADPKD is derived from that elaborated for CKD due to other causes. However, there are fundamental differences due to the peculiarities of the disease and the necessity to slow the growth of cysts.

For this reason, as the literature has demonstrated, it is fundamental to maintain a normal weight with a BMI between 18.5 and 24.9, since an overweight/obesity condition has been identified as a significant driver of cyst growth [43], resulting in a worsening of patients’ general condition, increasing progression of ESKD, and, consequently, worse fitness and QoL.

As precautionary measures, moderate coffee consumption and higher water intake are suggested, as this, in the same way as the decreased intake of salt/sodium may act, could decrease urinary osmolarity and, consequently, inhibition of AVP resulting in the suppression of cAMP [55].

Interesting prospects of nutritional treatment have been explored through the adoption of specific diets, such as calorie restriction, classical intermittent fasting, and some of its variants such as time-restricted feeding. The reason for studying these regimens is based on the inhibiting effect of calorie restriction on the mTOR pathway, which is constantly active in ADPKD patients leading to the inhibition of AMPK and stimulating cell proliferation and cystic development. As a result of the restriction, the ATP/AMP ratio decreases and AMPK is activated, resulting in mTOR inhibition. In experiments on animal models of PKD [45], it has been found that even with extremely moderate restriction (23% of daily food quantity), there are effects of slowing the cysts’ growth.

There is increasing interest in the ketogenic approach, analyzed in several animal model experiments. The rationale behind the use of this regime comes from the discovery of the metabolic inflexibility of cyst cells and their “Warburg-like” metabolism. Even the induction of a state of ketosis determines a state of nutrient deficiency that activates the inhibition pattern of mTOR. Moreover, the increase in circulating fatty acids leads to higher lipids uptake by the cyst cells, leading them to die from lipotoxicity. Another actor in this process seems to be BHB, the ketone body mainly produced in ketosis, which alone would modulate the pathway of mTOR and AMPK. In support of the ketogenic approach and integration with BHB, Torres et al. in a recent study hypothesized that the benefits seen with various intermittent fasting regimens are not due to kcal restriction by itself but are to be attributed to the partial ketosis induced by fasting itself, resulting in an increase in circulating ketone bodies, in particular BHB [54,55].

It is important to note that the majority of studies that analyze nutritional treatment on humans affected by ADPKD have the limitation of being observational studies, on small populations, and some of them have patient-reported measurements. Considering the impact of dietary regimens on the progression of the disease, it is desirable that a new randomized controlled trial would be developed to increase knowledge in this field.

As recommended by the KDIGO 2021 guidelines, in CKD conditions, the maintenance of at least 150 min of physical activity per week has been shown to have positive effects on overall health, counteracting functional decay and thereby reducing the risk of mortality [65]. In addition, it also has a positive effect on numerous cardiovascular risk factors and a positive effect in ADPKD on cyst growth [71].

In view of the great benefits to physical and mental health derived from physical activity, there is no doubt that people with ADPKD should lead an active lifestyle but with particular attention. For example, contact sports such as rugby, boxing, football, and others should not be practiced to prevent abdominal trauma from causing cyst rupture, resulting in bleeding and pain [4]. In addition, a further reason for a certain degree of precaution with regard to sport comes from recent studies [66,68,69], showing that patients with ADPKD have a particular cardiovascular profile and an impaired response to physical activity, already with normal GFR values.

The results from research underline the importance of a multidisciplinary approach to ADPKD patients in order to provide the best follow-up for patients with a disease that determines renal but also extrarenal manifestation.

Further studies are needed to increase the evidence currently present on the effect of diet and physical activity in polycystic patients and to understand which nutritional and behavioral strategies can best support current pharmacological strategies for slowing down kidney damage in ADPKD.

## Figures and Tables

**Figure 1 nutrients-15-02621-f001:**
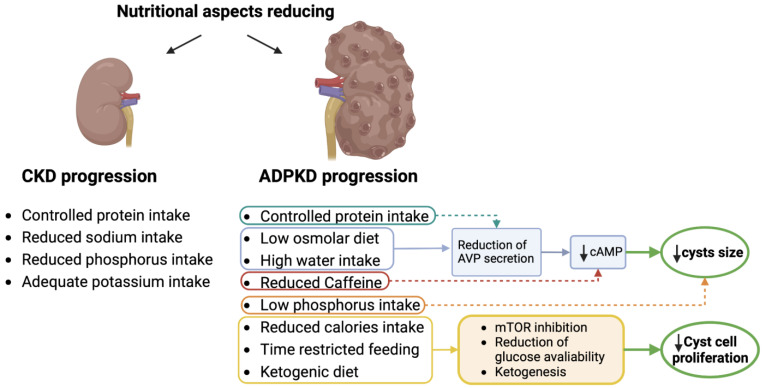
Impact of dietary approaches on the molecular pathways in ADPKD. (Cyclic adenosine monophosphate, cAMP; arginine vasopressin, AVP; mammalian target of rapamycin, mTOR).

**Table 1 nutrients-15-02621-t001:** Amount of protein intake in the CKD stages (EPI-CKD) [19].

CKD Stage	eGFR	Daily Amount of Protein
Stage 1	≥90 mL/min	0.8 mg/kg/die
Stage 2	60–89 mL/min	
Stage 3a	45–59 mL/min	
Stage 3b	30–44 mL/min	0.6 mg/kg/die
Sage 4	15–29 mL/min	0.6 mg/kg/die or 0.4–0.3 mg/kg/die + ketoanalogues
Stage 5	≤14 mL/min	

## Data Availability

No new data were created or analyzed in this study. Data sharing is not applicable to this article.

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
