# Peer review of "Diet and Physical Activity in Adult Dominant Polycystic Kidney Disease: A Review of the Literature"

_nutrients, 2023, doi:10.3390/nu15112621_

Round 1

Reviewer 1 Report

This article presents a comprehensive literature review on non-pharmacological therapeutic strategies for ADPKD, specifically focusing on diet and physical activity. It extensively explores the association of diet and physical activity with quality of life in ADPKD patients, highlighting the significance of dietary recommendations for ADPKD patients. The study is well designed and presented, however, there are some issues that need to be addressed:

1. The authors use both renal and kidney in the article. For example, phrases like “ inherited kidney disease”, chronic renal insufficiency,end stage renal disease are mentioned. However, according to the KDIGO consensus, it is recommended to replace the term renal with kidney instead. It might be better to use kidney throughout the whole article.

2. This study focused on the association of certain nutrients with ADPKD. Are there any literature that specifically investigated the impact of various dietary patterns, such as meditereanean diet, plant-based diet, DASH diet... on ADPKD?

3. It may be valuable to discuss potential limitations of the research and areas for future study, such as the need for more rigorous randomized controlled trials evaluating the efficacy of certain dietary interventions for ADPKD.

Reviewer 2 Report

The authors review the literature on the effect of diet and exercise on progression of ADPKD. They mention protein restriction, salt and phosphorus restriction, increased water intake calorie restriction and ketogenic diets and their review included the most recent literature. The conclusion is that none of these strategies is supported by enough evidence to implement it in clinical practice.
Major:

1. Ketogenic diet – at least in animal models – seems the most likely candidate for stabilisation of renal function in patients with ADPKD. On page 8, line 352ff the authors mention a study showing an increase in GFR in patients with ketosis of 7.3 ml/min per 1.73 m2. This could mean two things: a. Real improvement in renal function (if measurement of GFR was done by creatinine and/or urea-clearance or by scintigraphy). This however could be due to hyperfiltration and would not be a desirable effect.

b. If GFR were estimated as eGFRcreat, the increase in GFR would be due to decreased creatinine values, which could mean malnourishment in patients exposed to that diet.

c. Page 9 line 382ff: The same argument can be made for the study by Greenwood et al (#54) where creatinine dropped by 5.8%. In my view, a successful diet would not improve renal function but stabilize it.

2. Page 10 line 465: In their conclusion, the authors strongly encourage a higher water intake. While this most likely does not cause harm, a beneficial effect has also not been proven. A publication by Rangan et al (2022 NEJM DOI: 10.1056/EVIDoa2100021) on 182 patients with ADPKD showed a significant reduction in urine osmolality, no adverse effects but also no beneficial effect (total kidney volume, renal function) in the 92 patients with increased water consumption. This paper should also be mentioned in this review.
Minor:

1. Page 1 line 22: in-hibit

2. Page 3 line 135: Protein restriction to 0,6 g/kg/day is advised in CKD 3 and to 0,3-0,6 in CKD stage 4 and 5. While this can be done under strict control to avoid malnutrition most experts would not advise protein intake lower than 0.8 g/kg/day in the majority of patients (Dietary recommendations for patients with nondialysis chronic kidney disease - UpToDate).

3. Page 4 line 171: potassium target values are given as 3.5-5.5 mmol/l. To my knowledge mortality increases significantly at potassium levels of 5.5 and target values are given as 3,5-5.0 mmol/l.

4. Page 5 figure: Adeguate potassium intake – should be adequate

5. Page 11 line 498: “Fare clic o toccare qui per immettere il testo” ???

no  comments.
